# Prognostic Value of Red Blood Cell Distribution Width in Resected pN1 Lung Adenocarcinoma

**DOI:** 10.3390/cancers12123677

**Published:** 2020-12-08

**Authors:** Francesco Petrella, Monica Casiraghi, Davide Radice, Elena Prisciandaro, Stefania Rizzo, Lorenzo Spaggiari

**Affiliations:** 1Department of Thoracic Surgery, IRCCS European Institute of Oncology, 20141 Milan, Italy; monica.casiraghi@ieo.it (M.C.); elena.prisciandaro@ieo.it (E.P.); lorenzo.spaggiari@ieo.it (L.S.); 2Department of Oncology and Hemato-Oncology, Università degli Studi di Milano, 20141 Milan, Italy; 3Department of Biostatistcs, IRCCS European Institute of Oncology, 20141 Milan, Italy; davide.radice@ieo.it; 4Department of Radiology, Ente Ospedaliero Cantonale (EOC) Istituto di Imaging della Svizzera Italiana (IIMSI), 6903 Lugano, Switzerland; stefaniamariarita.rizzo@eoc.ch

**Keywords:** red blood cell distribution width (RDW), lung adenocarcinoma, disease-free interval

## Abstract

**Simple Summary:**

Red blood cell distribution width is a measure of the variation of erythrocyte volume. Impaired erythropoiesis can lead to a wide variation in erythrocyte dimension—defined as anisocytosis—indicating that pathological modifications are taking place. Recently, red blood cell distribution width has been advocated as an effective prognostic factor in cardiovascular diseases, acute kidney injury, autoimmune disease, and oncologic settings. In many advanced and several early-stage oncologic conditions, it has shown excellent prognostic efficacy; we therefore investigated what prognostic role red blood cell distribution width may have in resected lung cancer, focusing on pN1 adenocarcinoma patients in whom adjuvant treatments—although well-established—are still proposed case by case. Our findings suggest that red blood cell distribution width is strictly related to disease-free survival; it could therefore be considered as a further tool for planning postoperative adjuvant treatments and setting up an adequate follow-up program.

**Abstract:**

*Background:* Red blood cell distribution width is a measure of the variation of erythrocyte volume and has recently been advocated as a prognostic tool in neoplastic and non-neoplastic diseases. We studied the prognostic role of preoperative red blood cell distribution width (RDW) in resected pN1 lung adenocarcinoma patients. *Methods:* Sixty-seven consecutive pN1 lung adenocarcinoma patients operated in the last two years were retrospectively evaluated in the present study. Age, sex, smoking status, type of surgical resection, neoadjuvant and adjuvant treatments, pathological stage, T and N status, tumor size, preoperative hemoglobin (Hb) and RDW, preoperative neutrophils, lymphocytes, and their ratio were collected for each patient. Outpatient follow-up was performed and date of relapse was recorded. *Results:* There were 24 females (35.8%). Twenty-eight patients (41.8%) belonged to stage 3A and thirty-nine patients (58.2%) to stage 2B. Mean preoperative RDW % was 14.1 (IQR: 12.9–14.8). Univariate analysis disclosed preoperative RDW as strictly related to disease-free survival (*p* = 0.02), which was confirmed in the exploratory multivariable analysis (*p* = 0.003). *Conclusions:* Pre-operative RDW is an effective prognostic factor of disease-free survival in resected pN1 lung adenocarcinoma; it could therefore be considered as a further tool for planning postoperative adjuvant treatments and setting up an adequate follow-up program.

## 1. Introduction

Red blood cell distribution width (RDW) is a measure of the size of variation of circulating erythrocytes and it has recently emerged as one of the markers potentially implicated in the inflammatory process and oxidative stress as well as endothelial dysfunction in vessel diseases [1]. An increased variation in erythrocyte volume is defined as anisocytosis, a laboratory finding utilized for many years—almost exclusively—for clinical differential diagnosis of anemias [2].

More recently, RDW has been advocated as an effective prognostic factor in cardiovascular diseases [1], acute kidney injury [3], autoimmune disease [4], and oncologic settings [5] (Table 1).

In many advanced [6] and several early-stage oncologic conditions [7], RDW has shown excellent prognostic efficacy; we therefore investigated what prognostic role RDW may have in resected lung cancer, focusing on pN1 adenocarcinoma patients in whom adjuvant treatments—although well established—are still proposed case by case.

On one hand, in fact, the clinical practice guidelines of the European Society of Medical Oncology (ESMO) recommend adjuvant chemotherapy with a two-drug combination, preferably containing cisplatin; on the other hand, they clearly suggest considering pre-existing comorbidities, time since surgery, and postoperative recovery-need as well as making the decision through a multidisciplinary tumor board [8]. Offering adjuvant treatment in this group of patients may therefore sometimes be debatable and not always well defined.

Although N1 non-small cell lung—with the exclusion of stages T3 and T4—are classified as early-stage, many patients develop recurrence and die from cancer despite radical surgical resection; in particular, adenocarcinoma shows a significantly higher prevalence of recurrence when compared to other histo-types. The 5-year overall survival of patients with N1 stage II lung cancer ranges from 33% to 65%, thus suggesting that this is a heterogeneous subgroup of patients [8].

Complete blood counts (CBC) are routinely used in clinical practice in oncologic patients and their predictive value has been widely investigated in many solid tumors; several ratios (platelet to lymphocyte ratio, neutrophil to lymphocyte ratio, neutrophil to monocyte ratio) have emerged as significant prognostic factors [9,10,11].

It is well known that low hemoglobin values are associated with poor prognosis in many tumors, particularly lung cancer; similarly, it has been shown that RDW is associated with advanced tumor stages not only in the lung, but also in kidney and breast cancers [12,13,14]. Although it is not completely clear how RDW is clinically related with survival in cancer patients, it is believed that higher RDW values are triggered by folic acid, vitamin B12, and iron deficiencies due to malnutrition and chronic inflammatory conditions [15,16,17].

Similarly, the association between low hemoglobin levels and prognosis in oncologic patients may be explained by many conditions: mediator and cytokine release by neoplastic cells affecting hemoglobin production; suppression of erythropoietin and reduced response of progenitor cells to erythropoietin; and modification of the hematopoiesis environment resulting in reduced hemoglobin production [18].

Nowadays, there is clear evidence that inflammatory conditions—together with a weak immune system—are deeply involved in the growth, progression, and metastasis development of cancers; this may be due to many physical, chemical, and biological factors promoting cancer angiogenesis, worsening DNA damage, and favoring closer structure infiltration [19,20,21].

The aim of the present study was to investigate the prognostic role of preoperative RDW in resected pN1 pulmonary adenocarcinoma, to offer multidisciplinary teams an additional tool to better plan adjuvant treatment, or to set up a stricter post-operative follow-up program.

**Table 1 cancers-12-03677-t001:** Association between Red blood cell distribution width (RDW) and diseases in recent literature.

Author (ref. n.)	Year	RDW Relation with	Findings
Wang L [1]	2020	Ischemic stroke	RDW was an independent predictor of 3-month functional outcome, and a trend of dose-dependent relationship between RDW and 3-month death was detected.
Jia L [3]	2020	Acute kidney injury	RDW is positively correlated to survival time of 4-year follow-up in critically ill patients with acute kidney injury, and RDW is an independent prognostic factor of long-term outcomes of these patients.
Wang [6]	2020	Lung cancer	A higher value of pre-treatment RDW indicated worse survival of patients with lung cancer. RDW may serve as a reliable and economical marker for prediction of lung cancer prognosis
Toyokawa G [7]	2020	Lung cancer	RDW was shown to be associated with a worse long-term prognosis in resected pathologic stage I NSCLC patients
Seretis C [13]	2013	Breast cancer	RDW was significantly higher in patients with breast cancer, when compared to the enrolled patients with fibroadenomas.
Wang F [14]	2014	Renal cell carcinoma	RDW significantly higher in patients with renal cell carcinoma (RCC) than those in controls, and the baseline RDW was independently associated with RCC.
Clarke K [22]	2008	Inflammatory bowel disease	RDW is an effective differentiating test between Crohn’s disease and ulcerative colitis.
Li N [23]	2017	Cardio/cerebro vascular disease	Hypothetical and potential epidemiological associations between RDW and cardiovascular diseases.
Tonelli M [24]	2008	Coronary disease	Independent relation between higher levels of RDW and the risk of death and cardiovascular events in people with prior myocardial infarction.
Skjelbakken T [25]	2014	Myocardial infarction	RDW is associated with incident myocardial infarction in a general population independent of anemia and cardiovascular risk factors.
Zyczkowski M [26]	2017	Renal cell carcinoma	Cancer specific survival in patients receiving nephrectomy for renal cell carcinoma was significantly lower in patients with RDW ≥ 13.9%.
Qin Y [27]	2017	Ovarian cancer	RDW is associated with ovarian cancer and is a potential marker of its progression.
Chen GP [28]	2015	Esophageal cancer	RDW was an independent prognostic factor in patients with esophageal squamous cell carcinoma.
Wei TT [29]	2017	Gastric cancer	Patients with gastric cancer had significantly higher RDW than healthy controls.
Kemal Y [30]	2015	Endometrial cancer	Grade II and above endometrial cancer patients had higher levels of RDW than Grade I patients

## 2. Material and Methods

The present study was conducted in accordance with the Declaration of Helsinki [31] and was an observational retrospective study. Data were collected prospectively, entered into our institutional general thoracic database at the point of care, reviewed, and double-checked retrospectively.

Written informed consent to undergo the procedure and for the use of clinical and imaging data for scientific or educational purposes, or both, was obtained from all patients before the operation.

Each patient authorized the investigators to use their data anonymously only for scientific purposes according to Italian legislation (law no. 675/1996). Editors have been provided with a blank copy of the written informed consent.

Sixty-seven consecutive pN1 lung adenocarcinoma patients undergoing anatomical pulmonary resection during the last two years (from 2018 to 2019) were analyzed.

Standard lobectomy was defined as resection of only one lobe associated with radical mediastinal lymph node dissection, without any resection of mediastinal, chest wall, or diaphragmatic structures. Standard pneumonectomy was defined as the intrapericardial or extrapericardial removal of the entire lung associated with radical mediastinal lymph node dissection without any resection of mediastinal, chest wall, or diaphragmatic structures.

Operability was determined by standard clinical and radiographic procedures (whole-body computed tomography), nuclear imaging (whole-body fluorodeoxynucleotide positron emission tomography), and staging procedures including endobronchial ultrasonographic bronchoscopy and transbronchial needle aspiration, as appropriate. Amoxicillin and clavulanic acid were administered for the first five postoperative days in non-allergic patients, with the first dose administered before the skin incision. Thromboprophylaxis was maintained with sequential compression devices, early ambulation, and low-molecular-weight heparin delivered subcutaneously.

Age, sex, smoking status, type of surgical resection, neo-adjuvant and adjuvant treatments, pathological stage, T and N status, N1a and N1b status, tumor size, preoperative hemoglobin (Hb) and red blood cell distribution width (RDW), preoperative neutrophils and lymphocytes and their ratio were collected for each patient. Outpatient follow-up was performed in order to record the date of relapse (if any).

We stratified preoperative comorbidities according to an adapted Charlson comorbidity index [22] including a history of myocardial infarction, peripheral disease, cerebrovascular disease, diabetes (without end-organ damage), mild and moderate liver disease, and moderate kidney disease.

### Statistical Methods

Patients’ characteristics and blood parameters were summarized either by count and percentage or mean, standard deviation (SD), inter quartile range (IQR), median, and min and max for categorical and continuous variables, respectively. Univariate and multivariable competing risk (Fine–Grey model) disease-free survival analysis was performed, taking into account the competing nature of 30-days perioperative death with respect to relapse. The results of the Fine–Grey model are presented as both cumulative incidences of relapse and hazard ratios (HR) with 95% confidence intervals (95% CI). Disease-free survival was defined as the time from the date of surgery to relapse. Timing of patients with no evidence of disease and still alive at the end of the study was considered censored. Median follow-up time was computed by the reverse Kaplan–Meier method. All tests were two-tailed and considered significant at the 5% level. All analyses were done using SAS 9.4 (N.C., Cary, NC, USA).

## 3. Results

Mean age was 67.7 years (SD = 8.6) and there were 24 females (35.8%). Sixty patients (89.6%) were smokers or former smokers. Twenty-eight patients (41.8%) belonged to stage 3A and thirty-nine patients (58.2%) to stage 2B. Fifty-five patients (82.1%) were N1 single station (N1a) while 12 patients (17.9%) were N1 multiple stations (N1b). There were 11 pT1 patients (16.4%), 28 pT2 patients (41.8%), 16 pT3 patients (23.9%), and 12 pT4 patients (17.9%). Nine patients (13.4%) received neo-adjuvant treatments and 25 patients (37.1%) received adjuvant treatments. Fifty patients (74.6%) were operated on by total muscle sparing lateral thoracotomy and 17 patients (25.4%) by the minimally invasive approach (video-assisted thoracic surgery VATS or robot-assisted thoracic surgery RATS). Fifty-six patients (83.6%) received lobectomies and eleven patients (16.4%) received pneumonectomies (Table 2).

Mean preoperative laboratory values were: hemoglobin 13.3 g/dL (IQR 12.5–14.5); neutrophils 4.86 × 10^3^/µL (IQR: 3.60–5.62); lymphocytes 1.85 × 10^3^/µL (1.35–2.17); RDW 14.1(%) (IQR 12.9–14.8); and neutrophile/lymphocyte ratio (NLR) 3.05 (IQR 1.88–3.66) (Table 3). Overall, 16 (23.9%) patients relapsed within 22 months from the date of surgery, corresponding to a cumulative relapse incidence equal to 5.4% at six months and 93.6% at 22 months (Table 4 and Figure 1). Four (6.0%) patients died without signs of disease and entered the Fine–Grey model as competing events. No patients died of disease. Relapse was not significantly associated with any patient characteristics or blood parameters, except for RDW at both univariate (HR = 1.29, 95% CI: 1.04–1.59, *p* = 0.02) (Table 5) and multivariable (HR = 1.35, 95% CI: 1.11–1.65, *p* = 0.003) analyses (Table 6). Multivariable hazard ratio for RDW was adjusted for age and N1 status, showing a significant increased HR for N1b status vs. N1a (HR = 3.61, 95% CI: 1.36–9.58, *p* = 0.01). We did not observe any relationship between preoperative comorbidities and RDW.

We divided the population study into two groups based on that RDW cut-off value, which showed a significant difference for the outcome, then the hazard ratio for patients with a RDW value above the threshold value of 13.3% was HR = 6.23, 95% CI: (1.58, 24.6) (*p* = 0.009).

However, given the sample size and the research design, the search of such a threshold is sampling dependent.

## 4. Comments

RDW is a measure of the variation in the size of the circulating red blood cells and is defined as the coefficient of variation of the erythrocyte size [23]. Clinical disorders in which erythrocytes significantly vary in size are often due to impaired red blood cell production or augmented disruption [24]. Impaired erythropoiesis can eventually lead to a wide variation in erythrocyte dimension—defined as anisocytosis—indicating that some pathological modifications are taking place.

It has been widely reported by many epidemiological studies that patients suffering from cardiovascular diseases present anisocytosis more frequently due to higher RDW levels; however, the underlying pathophysiological pathway remains unclear [25,26,32].

More recently, several studies disclosed that RDW can be used as a prognostic marker in many different neoplastic diseases such as renal cell carcinoma [27], ovarian cancer [28], esophageal cancer [29], gastric cancer [30], endometrial cancer [33], breast cancer [34], and lung cancer [35].

Ichinose et al. reported higher RDW values as an independent risk factor for disease-free survival (DFS), overall survival (OS), and postoperative morbidity in elderly patients submitted to pulmonary resection for non-small cell lung cancer, but did not confirm this finding in younger patients [35]. On the contrary, our findings—although confirming that RDW is an independent risk factor for DFS—did not show any significant relationship with age, being effective both in young and elderly patients (Table 5).

The International Association for the Study of Lung Cancer recently suggested classifying N1 patients from low (pN1a) to high (pN1b) nodal tumor burden, disclosing that pN1 in only one region (hilar or lobar) was associated with a better outcome than neoplastic involvement of both nodal stations after surgery and adjuvant therapy [36,37]. These findings were confirmed by our results: in fact, our exploratory multivariable analysis showed that N1 status is probably an independent risk factor, with a significant higher risk for N1b patients (Table 5).

Our findings confirm that higher RDW values are significantly associated with a higher risk of a shorter DFS, while no conclusion can be drawn in terms of overall survival because of the relatively short follow-up period due to the only recent introduction of routine preoperative RDW in 2018; in fact, early post-operative deaths were competing events on relapse.

Although it is well known that higher RDW is related to unfavorable outcomes in many clinical disorders, the reason for this association is still unknown. As high RDW is related to systemic inflammation, impaired nutritional status, and bone marrow malfunction, it has been thought that it may be associated with impaired antineoplastic immunity [33].

We decided to test the prognostic value of preoperative RDW in this very selected group of patients (pN1) because—although clinical guidelines suggest adjuvant treatments [8]—they also recommend further evaluation of each patient in a multidisciplinary meeting.

In fact, in daily clinical activity, there are many patients in whom the advantages of additional treatments need to be carefully balanced with potential adverse events and major side effects. In the light of such a complex decisional process, we suggest that preoperative RDW may further contribute to a more accurate decision.

RDW is an effective prognostic factor not only in oncologic disease; it is in fact strictly related to the mortality of patients suffering from chronic obstructive pulmonary disease (COPD) and pulmonary hypertension (PH), both conditions very frequently associated with lung cancer.

Although higher RDW values are observed in patients with PH and COPD, the underlying pathway is not well-known; several studies have suggested that higher RDW levels in cardiovascular diseases with compromised lung function could be related to higher levels of oxidative stress and inflammation [38,39]. Hypoxemia related to airway obstruction in COPD patients may worsen oxidative stress, thus increasing both RDW and PH; moreover, chronic inflammation—observed both in neoplastic and non-neoplastic disease—increases the release of cytokines and mediators interfering with iron metabolism and bone marrow function, thus inhibiting hematopoiesis and increasing RDW values [38,39].

An elevated RDW implicates dysfunctional erythropoiesis, increased red blood cell destruction, or reduced red blood cell lifecycle; it has been reported that increased oxidative stress might be responsible for RDW level increase; this culminates in ineffective erythropoiesis due to chronic inflammation.

In addition, RDW is related to many inflammatory markers such as C-Reactive protein and several inflammatory cytokines (tumor necrosis factor-α, interleukin(IL)-6, (IL)-8, and IL-1β, affecting iron metabolism and bone marrow function); this environment inhibits erythropoiesis and the entry of larger and younger cells into the peripheral circulation, thereby increasing the RDW.

Increased RDW values have been found in pulmonary hypertension and chronic obstructive pulmonary diseases, both frequently diagnosed in lung cancer patients. With regard to our study population, given that all patients received careful preoperative assessment and were considered fit for major pulmonary resection, severe COPD or severe pulmonary hypertension were not diagnosed. We may therefore argue that these comorbidities could interfere with the prognosis of lung cancer patients, but their impact is probably less in very selected surgical patients.

RDW is also a simple, inexpensive laboratory test included in standard complete blood counts (CBC) routinely performed before surgery; therefore, it does not require any additional procedure, cost, or dedicated equipment; its cost-effectiveness, considering its valuable prognostic efficacy, represents an added value of the test.

A strict postoperative surveillance program may improve survival by diagnosing asymptomatic recurrences in patients with a better performance status, thus amenable to a more effective treatment; on the contrary, as the survival benefits and cost-effectiveness of postoperative surveillance is still unclear, preoperative RDW can be considered as a further tool to better assess the follow up program and select patients with a higher risk of recurrence [40].

Due to the small number of patients and events, the search for a threshold was strongly dependent on sampling and therefore unreliable and not generalizable. Vice versa, the inclusion of RDW in the model as a continuous variable allowed us not only to test its significance, but also its possible clinical relevance on the risk of relapse unrelated to any arbitrary and sampling-dependent threshold. Thus, based on our data using RDW as a continuous variable, we can still estimate the clinical relevance of RDW as an increased risk of relapse of approximately 35% for each 1% increase in the RDW value (see footnote in Table 6).

Park et al. demonstrated that in N1 lung adenocarcinoma patients, recurrences frequently occurred in the bones, lung, and brain, most often within the first three years after surgical resection; in many cases (about 40%), they observed an isolated recurrence that could benefit from local treatment. They therefore suggest routine brain, bone, and chest imaging during the first three years after surgery, especially for patients at a higher-risk of recurrence that can be better identified by using preoperative RDW [40].

The vast majority of our patients were staged clinical N1 before surgical resection and were therefore operated on by standard total muscle-sparing lateral thoracotomy; in fact, at our institution, we reserve the minimally invasive technique for N0 early-stage lung cancer. Seventeen patients operated by minimally invasive approach (video-assisted thoracic surgery VATS or robot-assisted thoracic surgery RATS) were clinical N0 and presented upstaging during the procedure due to micrometastasis to N1 stations.

In the present international staging system (TNM), N1 status is described as metastasis to peribronchial lymph nodes or hilar lymph nodes on the same side as the resected tumor, or intraparenchymal lymph-node metastasis due to direct infiltration of the tumor. It is postulated that lymphatic diffusion of neoplastic cells causes metastases to regional lymph nodes; in contrast, direct infiltration of intraparenchymal lymph-nodes may be due to tumor cell diffusion along the airway. The two different mechanisms can explain the different outcomes of these two types of lymph-node metastases [41,42].

Although monocyte to lymphocyte ratio (MLR) and platelet to lymphocyte ratio (PLR) have been described as effective prognostic factors in gastric and lung cancers, our findings did not disclose any significant relationship either with OS or with DFS [43,44].

Despite our interesting findings on the predictive value of preoperative RDW, we have to point out some limitations of the study: this was a retrospective, monocentric study with a limited number of patients, albeit focused on a strictly selected population; the small sample size was due to the only recent introduction of routine preoperative RDW in 2018. In addition, we were not able to evaluate the impact of RDW on overall survival because of the short duration of the follow-up.

## 5. Conclusions

Pre-operative RDW is an effective DFS prognostic factor in resected pN1 lung adenocarcinoma, therefore it could be considered as a further tool for better planning of adjuvant treatment or for setting up a strict post-operative follow-up program in this subset of patients.

## Figures and Tables

**Figure 1 cancers-12-03677-f001:**
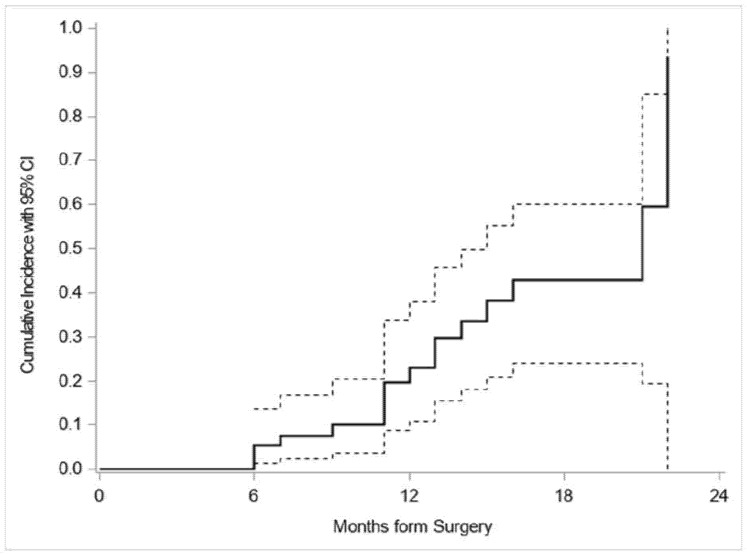
Cumulative incidence function for relapse after surgery.

**Table 2 cancers-12-03677-t002:** Patients’ demographic, treatment, and procedure summary statistics.

Characteristic	Level		Statistics *^a^*
Age at Surgery, years			67.7 (8.6)
Tumor Size, mm			46.5 (25.9)
Female Gender			24 (35.8)
Smoke	Yes		60 (89.6)
Surgery	Open		50 (74.6)
	MIS *^b^*		17 (25.4)
Pathological Stage	T1-2		39 (58.2)
	T3-4		28 (41.8)
N1 Status	N1A (single station)		55 (82.1)
	N1B (multiple stations)		12 (17.9)
Stage	3A		28 (41.8)
	2B		39 (58.2)
Grading	1		1 (1.7)
	2		20 (34.5)
	3		37 (63.8)
Adjuvant Treatments	No		42 (62.9)
	CT		18 (26.9)
	RT		5 (7.5)
	CT/RT		1 (1.5)
	CT/RT/Other		1 (1.5)
Neoadjuvant Treatments	No		58 (86.6)
	CT		9 (13.4)
Procedures	Upper Lobectomy	Left	18 (26.9)
		Right	17 (25.4)
	Lower Lobectomy	Left	6 (9.0)
		Right	9 (13.4)
	Pneumonectomy	Left	4 (6.0)
		Right	7 (10.5)
	Upper Sleeve Lobectomy	Left	1 (1.5)
		Right	1 (1.5)
	Lower bilobectomy		2 (3.0)
	Culminectomy		1 (1.5)
	Middle Lobectomy		1 (1.5)

*^a^* Mean (SD) for age and Tumor size, N (%) other variables; *^b^* Minimally invasive.

**Table 3 cancers-12-03677-t003:** Blood parameter summary statistics.

	Mean (IQR) *^a^*	Median (Min, Max)
Hemoglobin, g/dL	13.3 (12.5–14.5)	13.4 (9.1,16.1)
Neutrophils, ×10^3^/µL	4.86 (3.60–5.62)	4.41 (1.80,9.82)
Lymphocytes, ×10^3^/µL	1.85 (1.35–2.17)	1.69 (0.82,6.20)
RDW (%)	14.1 (12.9–14.8)	13.7 (11.3,18.9)
Neutrophil/Lymphocytes Ratio	3.05 (1.88–3.66)	2.70 (0.58,11.3)

*^a^* IQR: Inter Quartile Range

**Table 4 cancers-12-03677-t004:** Overall cumulative relapse incidence.

Months from Surgery	Patients Relapsed (Failed Events)	CIF *^a^* (%)
0	0	0
6	3	5.4
12	9	23.0
22	16	93.6

*^a^* CIF: Cumulative Incidence Function estimate; Total failed events, N = 16; Competing events (deaths), N = 4; Censored, N = 47

**Table 5 cancers-12-03677-t005:** Disease-free survival univariate analysis.

Risk Factor at Surgery		HR (95% CI)	*p*-Value
Age		0.98 *^a^* (0.75–1.27)	0.87
Tumor Size		1.06 *^b^* (0.86–1.30)	0.59
Hemoglobin		0.77 *^c^* (0.53–1.12)	0.17
Neutrophiles		1.08 *^c^* (0.83–1.39)	0.58
Lymphocytes		1.01 *^c^* (0.78–1.29)	0.96
Neutrophil/Lymphocytes Ratio		1.08 *^c^* (0.81–1.43)	0.62
RDW %		1.29 *^c^* (1.04–1.59)	0.02
N1 Status	N1A	1	
	N1B	2.63 (0.90–7.64)	0.08
Gender	Females	1	
	Males	0.86 (0.30–2.42)	0.77
Smoker	No	1	
	Yes	Not estimable	
Surgery	Open	1	
	MIS *^d^*	0.77 (0.27–2.19)	0.62
pT	1–2	1	
	3–4	1.83 (0.73-4.59)	0.20
Grading	1–2	1	
	3	1.21 (0.38–3.86)	0.75
Stage	3A	1	
	2B	0.55 (0.22–1.37)	0.55
Treatments	No	1	
	Yes	0.46 (0.13–1.61)	0.23
Procedure	Lobectomy	1	
	Pneumonectomy	0.39 (0.06–2.71)	0.34

*^a^* by 5-years increase; *^b^* by 10 mm increase; *^c^* by 1 unit increase; Median Follow-up = 10 months; *^d^* MIS: minimally invasive surgery

**Table 6 cancers-12-03677-t006:** Multivariable disease-free survival analysis.

Risk Factor at Surgery		HR (95% CI)	*p*-Value
Age at Surgery		0.97 *^a^* (0.75–1.26)	0.82
RDW %		1.35 *^b^* (1.11–1.65)	0.003
N1 Status	N1A	1	
	N1B	3.61 (1.36–9.58)	0.01

*^a^* by 5-years increase; *^b^* by 1% increase

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
