# Peer review of "Prognostic Value of Red Blood Cell Distribution Width in Resected pN1 Lung Adenocarcinoma"

_cancers, 2020, doi:10.3390/cancers12123677_

Round 1

Reviewer 1 Report

As a reviewer, I enjoyed reading this paper. The authors tried to identify red blood cell distribution width as a major prognostic factor after surgery for NSCLC with N1 mets.

Most weakness of this paper is that the sample size (67 patients in total) is very small. No generalized findings could be obtained nor inferred from this insufficient number of samples. I expect the authors to reconsider and modify the research design.

The authors did not clearly show the raw data of RDW, so the readers can’t tell the threshold value between favorable and unfavorable prognostic values. Thus, based on their data, this measurement cannot be applicable to the clinical settings, in terms of considering the indication for adjuvant chemotherapy or devising the most effective follow-up program in each patient.

RDW values are elevated in many pulmonary diseases, such as COPD and PH, which may suggest the high RDW in lung cancer may just reflect the confounding factors, leading to worse prognosis.

This paper lacks the description regarding the biological significance or molecular mechanisms of RDW affecting the cancer prognosis, which may undermine this paper.

The manuscript is far from the MDPI’s format, and the authors should at least follow the journal’s guideline.

Author Response

Reviewer 1

1) As a reviewer, I enjoyed reading this paper. The authors tried to identify red blood cell distribution width as a major prognostic factor after surgery for NSCLC with N1 mets. Most weakness of this paper is that the sample size (67 patients in total) is very small. No generalized findings could be obtained nor inferred from this insufficient number of samples. I expect the authors to reconsider and modify the research design.

 We fully agree on this major limit of the study (small sample size) and we have further highlighted this limit in the revised version; unfortunately this is due to the only recent introduction of routine preoperative RDW in 2018 at our Institution. This precludes  a wider sample size by expanding retrospectively our enrollment; on the other hand  we focused only on pN1 adenocarcinoma because of its peculiarity and present debate about post-operative treatments, that is not an issue in pN0 or in pN2 patients.

We added this information in the revised version  lines 271-272

“…we have to point out some limitations of the study: this is a retrospective, monocentric study with a limited number of patients, although focused on a strictly selected population; the small sample size is due  to the only recent introduction of routine preoperative RDW in 2018.”

2) The authors did not clearly show the raw data of RDW, so the readers can’t tell the threshold value between favorable and unfavorable prognostic values. Thus, based on their data, this measurement cannot be applicable to the clinical settings, in terms of considering the indication for adjuvant chemotherapy or devising the most effective follow-up program in each patient.

Due to the small number of patients and events, the search for a threshold was strongly dependent on sampling and therefore unreliable and not generalizable. Vice versa the inclusion of RDW in the model as a continuous variable has allowed us not only to test it significance but also its possible clinical relevance on the risk of relapse unrelated to any arbitrary and sampling-dependent threshold. Thus based on our data using RDW as a continuous variable we can still estimate the clinical relevance of RDW as an increased risk of  relapse of approximately 35%  for each 1% increase in the RDW value (see footnote in Table 6).

From the clinical point of view, we therefore would recommend a closer follow up in patients with preoperative RDW values  above the higher normal value  of  14.5 CV%

Revised version, lines 238 - 246

3) RDW values are elevated in many pulmonary diseases, such as COPD and PH, which may suggest the high RDW in lung cancer may just reflect the confounding factors, leading to worse prognosis.

We emphasized this aspect in the discussion of the revised version lines 223 - 228, arguing that  this impact can be more significant in non-surgical patients rather than in very selected patients undergoing major pulmonary resection, receiving a careful preoperative assessment excluding  severe preoperative comorbidities.

“Increased RDW values have been found in pulmonary hypertension and chronic obstructive pulmonary diseases, both frequently diagnosed in lung cancer patients. With regard to our populations study, given that all patients received careful preoperative assessment and  were considered fit for major pulmonary resection, severe COPD or severe pulmonary hypertension were not diagnosed; so we may argue that these comorbidities may interfere with lung cancer patients prognosis, but probably their impact is less in very selected  surgical patients.”

4) This paper lacks the description regarding the biological significance or molecular mechanisms of RDW affecting the cancer prognosis, which may undermine this paper.

 We added further possible explication of biological relation between RDW and cancer prognosis in the discussion section of the revised version, lines 215 – 222

“An elevated RDW implicates dysfunctional erythropoiesis, increased red blood cell destruction, or shortened red blood cell lifecycle; it has been reported that  increased oxidative stress might be responsible for RDW levels increase; this culminate in ineffective erythropoiesis due to chronic inflammation. Moreover RDW is related to many inflammatory markers, such as C- Reactive Protein and several  inflammatory cytokines (tumor necrosis factor-α, interleukin(IL)-6, (IL)-8, and IL-1β, affecting  iron metabolism and bone marrow function);  this environment  inhibits erythropoiesis and the entry of larger and younger  cells into the peripheral circulation, thereby increasing the RDW.”

5) The manuscript is far from the MDPI’s format, and the authors should at least follow the journal’s guideline.

Thanks to editor support, we now resubmit the paper in the appropriate format

Reviewer 2 Report

This observational retrospective study by Petrella and colleagues explores the prognostic value of red blood cell distribution (RDW) in patients with pN1 pulmonary adenocarcinoma. Data that were entered prospectively in the general thoracic database at the point of care for sixty-seven patients that underwent anatomical pulmonary resection in 2018 and 2019 were analyzed retrospectively. The authors found that disease free survival was significantly associated with RDW based on both a univariate and multivariate analysis. No other patient characteristics or blood parameters investigated in this study showed a significant association with disease relapse. These findings introduce the possibility of using RDW from a routinely used test in the clinic, CBC, to develop a treatment plan for patients with pN1 pulmonary adenocarcinoma (perhaps regardless of age, based on early indications from this study). Although this study has some limitations (a retrospective study, from a single center with a small number of patients) it is, nonetheless, valuable. A strength is the focus on a subset of patients that may benefit from these insights for improved treatment planning based on information that can be readily obtained with little additional cost from routine tests. This is study is interesting and will be of value to the clinical and scientific community. 

The manuscript is recommended for publication following some minor revisions:

  1. It might be interesting to see if other cell ratios (other than neutrophils/lymphocytes) showed similar lack of association with DFS.
  2. The manuscript has numerous errors related to grammar and spelling. These errors are distracting from an otherwise interesting manuscript. Although the challenges associated with having to publish in a non-native language are understandable, and unfortunate, the whole article will benefit greatly from a careful proofreading, preferably by an English expert.

Author Response

Reviewer 2

This observational retrospective study by Petrella and colleagues explores the prognostic value of red blood cell distribution (RDW) in patients with pN1 pulmonary adenocarcinoma. Data that were entered prospectively in the general thoracic database at the point of care for sixty-seven patients that underwent anatomical pulmonary resection in 2018 and 2019 were analyzed retrospectively. The authors found that disease free survival was significantly associated with RDW based on both a univariate and multivariate analysis. No other patient characteristics or blood parameters investigated in this study showed a significant association with disease relapse. These findings introduce the possibility of using RDW from a routinely used test in the clinic, CBC, to develop a treatment plan for patients with pN1 pulmonary adenocarcinoma (perhaps regardless of age, based on early indications from this study). Although this study has some limitations (a retrospective study, from a single center with a small number of patients) it is, nonetheless, valuable. A strength is the focus on a subset of patients that may benefit from these insights for improved treatment planning based on information that can be readily obtained with little additional cost from routine tests. This is study is interesting and will be of value to the clinical and scientific community.

The manuscript is recommended for publication following some minor revisions:

1) It might be interesting to see if other cell ratios (other than neutrophils/lymphocytes) showed similar lack of association with DFS.

We explored monocyte to lymphocyte ratio (MLR) and platelet to lymphocyte ratio (PLR) and both did not disclose any significant relation neither with OS, nor with DFS. We added this information in the revised version  lines 266 – 268 and we added two more references (42 and 43).

“Although monocyte to lymphocyte ratio (MLR) and platelet to lymphocyte ratio (PLR) have been described as effective prognostic factors in gastric and lung cancers, our findings did not disclose any significant relation neither with OS, nor with DFS [42,43].”

  1. Chen L , Hao Y, Zhu L Monocyte to lymphocyte ratio predicts survival in patients with advanced gastric cancer undergoing neoadjuvant chemotherapy Onco Targets Ther      2017 Aug 10;10:4007-4016
  2. Diem S , Schmid S , Krapf M et al Neutrophil-to-Lymphocyte ratio (NLR) and Platelet-to-Lymphocyte ratio (PLR) as prognostic markers in patients with non-small cell lung cancer (NSCLC) treated with nivolumab   Lung Cancer 2017 Sep;111:176-181.

2) The manuscript has numerous errors related to grammar and spelling. These errors are distracting from an otherwise interesting manuscript. Although the challenges associated with having to publish in a non-native language are understandable, and unfortunate, the whole article will benefit greatly from a careful proofreading, preferably by an English expert

An English  native-speaker has now proofread the revised paper

Reviewer 3 Report

In this work, the authors investigate the prognostic role red blood cell distribution width may have in resected lung cancer. As authors mentions that ed blood cell distribution width can be an indicator of many pathological conditions, so for the analysis, it is not clear how the authors distinguish the impact of lung cancer from other diseases that could also lead to the increased width of RBC size distribution. For example. a recent study "Association of Red Blood Cell Distribution Width With Mortality Risk in Hospitalized Adults With SARS-CoV-2 Infection" showed the measurements from COVID-19 patient also show a  wider range of RBC size distribution.  Particularly, the patient cohort is mostly seniors who are likely to carry cardiovasicular diseases and diabetes. 

  1. In the introduction, it is better to have a table summarizing recent work on the association between RDF and diseases and the possible cause of the correlation.
  2. Other patient information such as drug medication, alcoholic and diabetes could also be considered in the analysis.

Author Response

Reviewer 3

In this work, the authors investigate the prognostic role red blood cell distribution width may have in resected lung cancer. As authors mentions that ed blood cell distribution width can be an indicator of many pathological conditions, so for the analysis, it is not clear how the authors distinguish the impact of lung cancer from other diseases that could also lead to the increased width of RBC size distribution. For example. a recent study "Association of Red Blood Cell Distribution Width With Mortality Risk in Hospitalized Adults With SARS-CoV-2 Infection" showed the measurements from COVID-19 patient also show a  wider range of RBC size distribution.  Particularly, the patient cohort is mostly seniors who are likely to carry cardiovasicular diseases and diabetes.

1) In the introduction, it is better to have a table summarizing recent work on the association between RDF and diseases and the possible cause of the correlation.

We added a new Table 1 summarizing recent work on the association between RDW and diseases and reporting each finding (very often in the paper is not postulated any possible cause of correlation). The other tables were then renumbered as Table 2,3,4,5 and 6.

2) Other patient information such as drug medication, alcoholic and diabetes could also be considered in the analysis.

In the revised version #1, we stratified preoperative  comorbidities according to an adapted Charlson comorbidity index  including  a history of myocardial infarction, peripheral disease, cerebrovascular disease, diabetes (without end-organ damage), mild and moderate liver disease, and moderate kidney disease. We added a further reference e we did not find any correlation between comorbidties and pre operative RDW.

Revised version lines 114-117 and lines 153,154.

New reference 23. M.E. Charlson, P. Pompei, K.L. Ales, C.R. Mackenzie A new method of classifying prognostic comorbidity in longitudinal studies: development and validation J Chronic Dis, 40 (1987), pp. 373-383

Round 2

Reviewer 1 Report

To the author,

I reviewed the paper very carefully and found little improvement on my concerns in the revised version. The sample size (67 patients in total) is still very small, and it was not solved in this revision round. No generalized findings could be obtained nor inferred from this insufficient number of samples. Besides, if the author would like to demonstrate the utility of RDW as a prognostic factor, the authors should divide the patients into two groups by the threshold value and compare the outcome. The research design is poor, thus the research is not interesting nor sound scientifically. It is not relevant to the clinical settings nor helpful for the readers. I expect the authors to reconsider and modify the research design.

Author Response

 x

Reviewer 3 Report

I can not locate the new table I in the revised manuscript and the number of the table in the draft is a mess.  Please check the manuscript carefully before resubmission.

Author Response

.
